# Differential Infectivity of Original and Delta Variants of SARS-CoV-2 in Children Compared to Adults

Lauren Garnett,[a,b] Carmen Tse,[c] Duane Funk,[d] Kerry Dust,[e] Kaylie N. Tran,[a] Adam Hedley,[b,e] Guillaume Poliquin,[a,f] Jared Bullard,[e,f] James E. Strong[a,b,f]

[a]Special Pathogens Program, National Microbiology Laboratory, Public Health Agency of Canada, Winnipeg, Manitoba, Canada
[b]Department of Medical Microbiology and Infectious Diseases, University of Manitoba, Winnipeg, Manitoba, Canada
[c]Department of Internal Medicine, University of Manitoba, Winnipeg, Manitoba, Canada
[d]Departments of Anaesthesiology and Medicine, Section of Critical Care, University of Manitoba, Winnipeg, Manitoba, Canada
[e]Cadham Provincial Laboratory, Manitoba Health, Winnipeg, Manitoba, Canada
[f]Department of Pediatrics & Child Health, University of Manitoba, Winnipeg, Manitoba, Canada

**ABSTRACT** Although children of all ages are susceptible to severe acute respiratory syndrome coronavirus 2 (SARS-CoV-2) infection, they have not been implicated as major drivers of transmission thus far. However, it is still unknown if this finding holds true with new variants of concern (VOC), such as Delta (B.1.617.2). This study aimed to examine differences in both viral RNA (as measured by cycle threshold [$C_T$]) and viable-virus levels from children infected with Delta and those infected with original variants (OV). Furthermore, we aimed to compare the pediatric population infection trends to those in adults. We obtained 690 SARS-CoV-2 RT-PCR positive nasopharyngeal swabs from across Manitoba, Canada, which were further screened for mutations characteristic of VOC. Aliquots of sample were then provided for $TCID_{50}$ (50% tissue culture infective dose) assays to determine infectious titers. Using a variety of statistical analyses we compared $C_T$ and infectivity of VOC in different age demographics. Comparing 122 Delta- to 175 OV-positive nasopharyngeal swab samples from children, we found that those infected with Delta are 2.7 times more likely to produce viable SARS-CoV-2 with higher titers (in $TCID_{50}$ per milliliter), regardless of viral RNA levels. Moreover, comparing the pediatric samples to 130 OV- and 263 Delta-positive samples from adults, we found only that the Delta pediatric culture-positive samples had titers ($TCID_{50}$ per milliliter) similar to those of culture-positive adult samples.

**IMPORTANCE** These important findings show that children may play a larger role in viral transmission of Delta than for previously circulating SARS-CoV-2 variants. Additionally, they may suggest a mechanism for why Delta has evolved to be the predominant circulating variant.

**KEYWORDS** COVID-19, Delta variant, RT-PCR, SARS-CoV-2, $TCID_{50}$

Severe acute respiratory syndrome coronavirus 2 (SARS-CoV-2), the etiological agent of coronavirus disease 2019 (COVID-19), has caused over 556 million cases and 6.3 million deaths globally as of 15 July 2022 (WHO dashboard). It has been previously reported that children of all ages are susceptible to SARS-CoV-2 infection; however, unlike for other respiratory viruses, they seem to account for fewer cases and deaths, milder symptoms, and a reduced role in viral transmission relative to adults (1–7). Nevertheless, with new SARS-CoV-2 variants of concern (VOC) rapidly appearing, it is important to understand and monitor how children contribute to ongoing infection and transmission trends. This is particularly imperative with reduced vaccine eligibility and the return of in-person learning at school and early-care settings, as the proximity

Address correspondence to James E. Strong, jim.strong@phac-aspc.gc.ca.

The authors declare no conflict of interest.

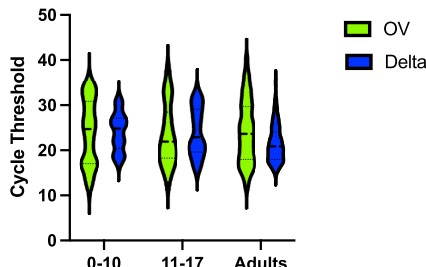

**FIG 1** Comparison of SARS-CoV-2 E gene RT-PCR $C_T$ value by VOC versus age group. Median $C_T$ values were higher for the Delta variant in pediatric age groups than in adults ($P < 0.001$; Kruskal-Wallis ANOVA), implying lower viral RNA levels. Median [IQR] $C_T$ was similar for children 0 to 17 years old between OV and Delta variant (24 [18 to 30] versus 23 [20 to 28]; $P = 0.97$).

of children to each other, educators, household contacts, and the wider community could lead to rapid transmission (8).

The SARS-CoV-2 Delta (B.1.617.2) variant first appeared with devastating effects in India and was quickly detected around the globe, causing surges in case numbers and subsequently becoming the predominant circulating strain by June 2021 (9). Delta was determined to be a VOC by the World Health Organization due to evidence of higher transmissibility, worse clinical implications, and decreased vaccine efficacy (10–13). Specifically, early data showed that Delta has a 60% increase in transmissibility in adults over previous VOC, including Alpha (B.1.1.7) (14).

Coinciding with the circulation of Delta, pediatric cases, hospitalizations, and COVID-19-related complications and deaths have also increased (15). This was evident from the Centers for Disease Control and Prevention work reporting up to a 10-fold increase in COVID-19-associated hospitalization rates (16). However, no studies specifically evaluating the infectivity of children infected with Delta have been completed. Therefore, the aim of this study was to investigate if there are any significant differences in cycle threshold ($C_T$) values or levels of viable infectious virus obtained from swab samples from children and adults infected with Delta and compare these rates to those of previously circulating original variants (OV).

## RESULTS

A total of 130 OV- and 263 Delta-positive samples from adults were compared to 175 OV- and 122 Delta-positive samples from children. In children, there was no difference in median cycle threshold ($C_T$) between OV and Delta samples (24 [18 to 30] versus 23 [20 to 28]; $P = 0.97$) (Fig. 1). However, the odds of cell culture positivity were significantly higher for pediatric Delta samples than for OV samples (odds ratio [OR], 2.7; 95% confidence interval [CI], 1.6 to 4.5; $P < 0.001$). We found viable virus in 41.0% (95% CI, 32.1 to 50.2%) of Delta samples compared to 20.5% (95% CI, 14.8 to 27.3%) of OV samples. In samples that were culture positive, the quantity of viable virus, measured as 50% tissue culture infective doses ($TCID_{50}$) per milliliter, was also significantly higher for Delta than for OV (5.62E+03 $TCID_{50}$/mL [5.62E+2 to 1.78E+04] versus 5.62E+02 $TCID_{50}$/mL [3.16E+02 to 3.16E+03]; $P < 0.001$) (Fig. 2).

When the pediatric samples were compared to the adult samples, children had a significantly higher mean $C_T$, denoting less viral RNA, for the Delta VOC (23.3 [19.8 to 27.6] versus 20.9 [18.0 to 24.1]; $P < 0.001$) (Fig. 1). Children also had a 0.55-lower odds ratio of a positive culture (95% CI, 0.36 to 0.85; $P = 0.008$) than adults. However, pediatric Delta samples that were culture positive resulted in similar $TCID_{50}$/mL titers compared to adult samples (5.62E+03 [5.62E+02 to 1.78E+04] versus 5.62E+03 [5.62E+02 to 3.16E+04], $P = 0.62$) (Fig. 2).

$C_T$ values for the beta globin (BGB) gene showed no statistical difference between samples by age (Fig. 3), thereby signifying that the quality of the samples was comparable between age groups. This also confirms that there were no differential PCR inhibitors

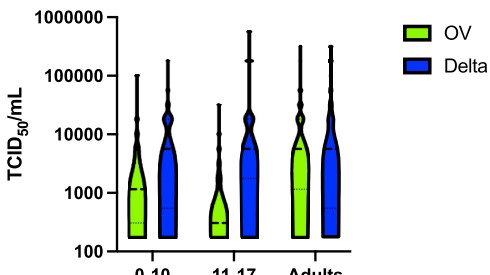

**FIG 2** Comparison of $TCID_{50}$ per milliliter by VOC versus age group. There was no difference in numbers of $TCID_{50}$ per milliliter between the different age categories with respect to the Delta variant specifically ($P = 0.68$; Kruskal-Wallis ANOVA). In the pediatric age groups, the $TCID_{50}$/mL value was significantly higher for the Delta variant in the 11- to 17-year age category than for OV (5,620 [1,780 to 17,800] versus 316 [178 to 2,125]; $P < 0.001$), but there was no difference in the 0- to 10-year age category.

within the samples as tested and that all reagents and protocols were working appropriately.

## DISCUSSION

**Interpretation.** Within the pediatric population, we found that individuals infected with Delta are 2.7 times more likely to produce viable SARS-CoV-2, and at higher titers, than age-matched individuals infected with OV. This important finding shows that children may play a larger role in viral transmission of Delta than previously circulating SARS-CoV-2 variants. Children and youth contributing to increased viral transmission of Delta may help explain one mechanism of how Delta became the predominant circulating variant (9). However, in agreeance with our previous study, children infected with Delta still have 45% lower odds of having a culture-positive sample than adults (17).

Children have borne a significant burden from disruptions in developmental, social, and educational activities from the SARS-CoV-2 pandemic, while their role in the spread of disease has been continuously scrutinized. Although earlier studies suggested that the spread of disease from and among children is low, more recent evidence suggests that this may be changing (15, 18). As highly transmissible VOC, such as Delta, arise globally, there is a renewed concern regarding the pediatric population's role in seeding outbreaks and an increasing importance to understanding the infection dynamics in this population (2, 19). In the absence of epidemiological evidence of the spread of the various variants of SARS-CoV-2 among and from children, we investigated differences in both $C_T$ and viable virus levels in children infected with Delta compared to OV. We then further compared the pediatric population infection trends to the adult population.

The physiological mechanism of age discrepancies for SARS-CoV-2 infection is still widely under investigation due to the potentially large number of biological, host, and environmental factors (20). These factors may include differences in the dynamics of shedding from children compared to adults and/or reflect differences in maturity of their respective immune systems (21).

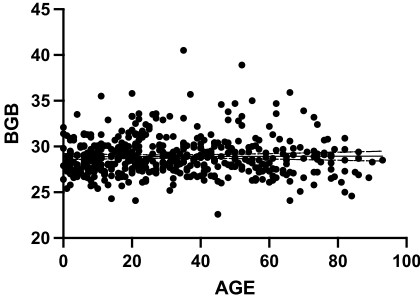

**FIG 3** Linear regression analysis of swab beta globin (BGB) gene $C_T$ versus age shows no statistical difference between BGB $C_T$ level and age at which the sample was obtained.

While it has been generally accepted that higher secondary attack rates are associated with higher viral loads, as measured by quantitative nucleic acid amplification methods (i.e., real-time PCR [RT-PCR]), others have shown that nucleic acid detection does not necessarily coincide with the presence of infectious virus (17, 22). The latter point is apparent in the results shown here, where $C_T$ levels were similar in children infected with Delta and OV; however, Delta-positive samples produced much higher culture positivity than previous variants. Here, culture positivity was evaluated using $TCID_{50}$/mL titers on cells as a surrogate for transmissibility, a much more specific test of infectivity than RT-PCR. This difference between Delta and OV culture positivity in children may reveal one aspect of its transmissibility advantage (23).

**Limitations.** There are limitations of this study that should be considered. These include the lack of clinical and epidemiological data linked to samples from this study. Incorporation of these data is an important next step in establishing SARS-CoV2 Delta VOC transmission dynamics in children. Additionally, without longitudinal sampling, any difference in onset and duration of viral shedding between Delta and OV in children cannot be determined. The duration of infectivity likely plays a major role in increased viral transmission of Delta compared to OV and should be further investigated. It is also possible that differences in RT-PCR and culturable virus between children and adults are accounted for by the nature or quality of the swab sampling differences between these demographics. However, the fact that the levels of the housekeeping gene (BGB) from adults and children are comparable makes this less likely (Fig. 3). Furthermore, the same samples are used for PCR testing and culturable virus, and so, if this is an accepted critique, we should not consider similar $C_T$ cutoffs in adults and children.

**Conclusion.** The relationship between age, SARS-CoV-2 viral load, and transmission has not been comprehensively explored and requires further investigation. Continuation of this work also involves monitoring of viral dynamics for new emerging SARS-CoV-2 VOC, such as Omicron (B.1.1.529), in different age populations. However, the results presented here, showing that Delta variant grows more often and to higher levels from nasopharyngeal swabs taken from children than previous variants of SARS-CoV-2, may have revealed changing infection trends within the pediatric demographic with novel VOC.

## MATERIALS AND METHODS

**Sampling collection.** Six hundred ninety nasopharyngeal swab samples from unvaccinated patients across Manitoba, Canada (population, 1.4 million), from March to December 2020 and July to September 2021 were provided to Cadham Provincial Laboratory, the public health reference laboratory for Manitoba, as part of routine clinical care and contact tracing. The study was performed in accordance with protocol HS23906 (H2020:211) as approved by the University of Manitoba Research Ethics Board.

**SARS-CoV-2 RT-PCR.** Samples positive for the SARS-CoV-2 envelope (E) gene by real-time PCR (RT-PCR) were further screened for mutations characteristic of VOC. The screening targets included spike region deletions in amino acids 69 and 70 and single nucleotide polymorphisms (SNP), including N501Y, E484K, and L452R. If quality and quantity of sample allowed, lineages were further confirmed by whole-genome sequencing using the Freed primer scheme (24). In addition, levels of the beta globin (BGB) gene, a standard housekeeping gene, were used to test the quality of the swab sample and as an nucleic acid integrity control as measured by RT-PCR (25).

**$TCID_{50}$.** Aliquots of patient samples were provided to the National Microbiology Laboratory for 50% tissue culture infective dose ($TCID_{50}$) determination. For $TCID_{50}$ evaluation, nasopharyngeal samples were diluted 10-fold and incubated on Vero cells (ATCC CCL-81) maintained in modified Eagle's medium (MEM) supplemented with 2% fetal bovine serum (FBS), 1% penicillin/streptomycin, and 1% L-glutamine at 37°C with 5% carbon dioxide for 96 h. Following incubation, cytopathic effect was evaluated under a microscope and recorded. $TCID_{50}$ and $TCID_{50}$ per milliliter were calculated using the Reed and Muench method as previously described (26).

**Statistical analysis.** Statistical analysis was carried out as previously described (17). Briefly, here we present normally distributed data with means and standard deviations and present nonnormally distributed data with medians and interquartile ranges (IQRs). We assessed normality using the Kolmogorov-Smirnov test. We performed between-group comparisons using Student's $t$ test or the Mann-Whitney test and used the Fisher exact test for categorical data. We compared nonparametric group medians using Kruskal-Wallis analysis of variance (ANOVA). We considered two-tailed $P$ values less than 0.05 significant. Statistical results are reported as (median [interquartile range]; $P$ value). We performed statistical analysis with Stata version 16.1 and GraphPad Prism 9. In our previous work, we found that adults had a culture positivity rate of 28.9% (27). Based on published data that the Delta variant may be 40 to 60% more infectious than wild-type SARS-CoV-2, and assuming that this value would be related to culture

positivity, we would require 116 pediatric samples to detect a 40% increase in culture positivity rates, with a power of 0.8 and $\alpha$ of 0.05 among children (28, 29).

## ACKNOWLEDGMENTS

We have no conflicts of interest to disclose.

No funding was secured for this study.

J.B., K.D., D.F., J.E.S., and G.P. conceptualized and designed the study, analyzed the data, and reviewed and revised the manuscript. C.T. and L.G. drafted the initial manuscript and performed revisions. A.H. collected, analyzed by RT-PCR, stored, and transported respiratory samples. He also reviewed and revised the manuscript. K.N.T., L.G., and J.E.S. collected, stored, and transported specimens in addition to performing cell culture experiments. They also reviewed and revised the manuscript. All authors approved the final manuscript as submitted and agree to be accountable for all aspects of the work.

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
