## [Reviewer comments · Microbiology Spectrum]

Microbiology Spectrum

Differential infectivity of original and Delta variants of SARS-CoV-2 in children compared to adults

Lauren Garnett, Carmen Tse, Duane Funk, Kerry Dust, Kaylie Tran, Adam Hedley, Guillaume Poliquin, Jared Bullard, and James Strong

Corresponding Author(s): James Strong, Public Health Agency of Canada

Review Timeline:

Submission Date:	February 3, 2022
Editorial Decision:	July 14, 2022
Revision Received:	July 27, 2022
Accepted:	August 1, 2022

Editor: Alison Sinclair

Reviewer(s): Disclosure of reviewer identity is with reference to reviewer comments included in decision letter(s). The following individuals involved in review of your submission have agreed to reveal their identity: ASAAD MOHAMMED ATAA (Reviewer #3)

Transaction Report:

DOI: <https://doi.org/10.1128/spectrum.00395-22>

July 14, 2022

Dr. James Eric Strong
Public Health Agency of Canada
Division of Special Pathogens
1015 Arlington Street
Winnipeg, Manitoba R3E 3P6
Canada

Re: Spectrum00395-22 (Differential infectivity of original and Delta variants of SARS-CoV-2 in children compared to adults)

Dear Dr. James Eric Strong:

Thank you for submitting your manuscript to Microbiology Spectrum.

As Editor I agree with the reviewers' comments, and I would welcome a revised manuscript that addresses all of these.

Link Not Available

Sincerely,

Alison Sinclair

Journals Department
Reviewer comments:

Reviewer #1 (Comments for the Author):

I appreciate the opportunity to review this interesting manuscript by Garnett et al which aimed to examine differences in both viral RNA and viable virus levels from children infected with Delta compared to original variants and to same measurements in adults. They compared Ct values and infectivity of VOCs in different age demographics. The study showed that children infected with Delta are 2.7 times more likely to produce viable SARS-CoV-2 with higher TCID50/mL titers, regardless of viral RNA levels

than those infected with original variants.

This manuscript has both epidemiological and virological importance but the manuscript requires some corrections before considering its publication.

Areas for revision and correction:

Please delete subheadings labelling numbers in the manuscript e.g. 1.0. Introduction

Abstract:

The abstract is concise and explains the study well. However to be in line with the journal style, replace the heading "Interpretation" with "Importance".

Methods:

SARS-CoV-2 RT-PCR

For the variant mutations, they were mentioned "N501Y, 484K and 452R", please write them consistently. E.g E484K etc

Statistical analysis:

Page 6, line 180 -184: There should be citation of your previous work on this subject and citation of other studies where you mentioned about the delta variant infectious rate. In addition, what method was used to calculate the sample size, please elaborate on this "In our previous work, we found that adults had a culture positivity rate of 28.9%. Based on published data that the Delta variant may be 40-60% more infectious than wild type SARS-CoV-2, and assuming that this value would be related to culture positivity, we would require 116 pediatric samples to detect a 40% increase in culture positive rates, with a power of 0.8 and α of 0.05 among children."

Figures

Write a figure legend page and the next pages put the figures without captions

In page 1, line 377: Figure 1 please explain what does these numbers mean? "(24, [18-30] vs. 23, [20377 28], P 0.97)."

In page 14, line 382: Figure 2 please write p-value as $p= 0.68$

In page 14, line 384: explain what these numbers are?

"Figure 2: Comparison of Tissue Culture Infective dose 50% (TCID50) per mL by variant of concern (VOC) vs. age group. There was no difference in TCID50/ml between the different age categories with respect to the Delta variant specifically (P 0.68 Kruskal-Wallis ANOVA). In the pediatric age groups, the TCID50/ml was significantly higher for the Delta variant in the 11-17 age category than for OV (5620 [1780-17800] vs. 316 [178-2125], P <0.001), but there was no difference in the 0-10 age category."

Reviewer #3 (Comments for the Author):

I would like to thank you for this wonderful work, which will be a qualitative addition to understanding important molecules of the COVID-19 pandemic, but there are some comments that should be reviewed.

In lines (118-120), You need updated statistical data for Coronavirus disease (COVID-19) pandemic.

In lines (239-243) It is preferable to delete this sentence. We still do not know what the side effects of vaccines are. Also, some studies have confirmed that there are side effects to the vaccine, and there are many opponents to the vaccine. Also, there are many voices that have not agreed to give vaccinations to children. The results of this study will tell us important things about how new mutations spread and how to stop them, and you don't need to mention the vaccine at the moment to children.

In References, You need to standardize the format for all references (style). There are some errors in the year of publication, as well as the contents of some references.

Staff Comments:

Preparing Revision Guidelines

For complete guidelines on revision requirements, please see the journal Submission and Review Process requirements at

<https://journals.asm.org/journal/Spectrum/submission-review-process>. **Submissions of a paper that does not conform to Microbiology Spectrum guidelines will delay acceptance of your manuscript. "**

Please return the manuscript within 60 days; if you cannot complete the modification within this time period, please contact me. If you do not wish to modify the manuscript and prefer to submit it to another journal, please notify me of your decision immediately so that the manuscript may be formally withdrawn from consideration by Microbiology Spectrum.

**Research: Observational Study**

**Differential infectivity of original and Delta variants of SARS-CoV-2 in children compared**
**to adults**

Lauren Garnett BSc^{1,2} Carmen Tse MD³, Duane Funk MD⁴, Kerry Dust PhD⁵, Kaylie N Tran
BSc¹, Adam Hedley BSc⁵, Guillaume Poliquin MD, PhD^{1,6} and Jared Bullard MD^{5,6}, James E.
Strong MD, PhD^{1,6}

**Affiliations:** ¹National Microbiology Laboratory, Public Health Agency of Canada, Winnipeg,
Manitoba, Canada, ²Department of Medical Microbiology and Infectious Diseases, University of
Manitoba, Winnipeg, Canada, ³Department of Internal Medicine, University of Manitoba,
Winnipeg, Manitoba, Canada, ⁴Departments of Anaesthesiology and Medicine, Section of
Critical Care, University of Manitoba, Winnipeg, Manitoba, Canada, ⁵Cadham Provincial
Laboratory, Manitoba Health, Winnipeg, Manitoba, Canada, ⁶Department of Pediatrics & Child
Health, University of Manitoba, Winnipeg, Manitoba, Canada

**Address correspondence to:** James E Strong, Public Health Agency of Canada - National
Microbiology Laboratory, 1015 Arlington Street, Winnipeg, Manitoba, Canada, R3E 3P6,
jim.strong@phac-aspc.gc.ca, 204-789-7032

**Short title: Delta VOC SARS-CoV-2 infectivity in children vs adults**

**Conflict of Interest Disclosures (includes financial disclosures):** The authors have no conflicts
of interest to disclose.

**Funding/Support:** No funding was secured for this study.

**Abbreviations:** Severe acute respiratory syndrome coronavirus (SARS-CoV-2), coronavirus
disease 2019 (COVID-19), variants of concern (VOC), reverse transcriptase polymerase chain
reaction (RT-PCR), cycle threshold (Ct), 50% tissue culture infective dose (TCID₅₀/mL)

**Word Count: 1802**

**Abstract word count: 249**

Contributor’s Statement Page

[revised manuscript text omitted]

24.1] P <0.001, Figure 1). Comparably, children also had a 0.55 reduction in the odds ratio of a
positive culture (95% CI 0.36-0.85, P 0.008) compared to adults. However, pediatric Delta
samples that were culture positive resulted in similar TCID₅₀/ml titres compared to adult samples
(5.62E+03 [5.62E+02 - 1.78E+04] vs. 5.62E+03 [5.62E+02 – 3.16E+04], P 0.62, Figure 2).

BGB Ct levels show no statistical difference between samples by age (Figure 3), thereby
signifying that the quality of the samples is comparable between age groups. This also confirms
that there are no differential PCR inhibitors within the samples as tested and that all reagents and
protocols are working appropriately.

**4.0.Discussion**

**4.1. Interpretation**

Within the pediatric population, we found that individuals infected with Delta are 2.7
209 times more likely to produce viable SARS-CoV-2 and at higher titres, compared to age matched
individuals infected with OVs. This important finding shows that children may play a larger role
in viral transmission of Delta than previously circulating SARS-CoV-2 variants. Children and
youth contributing to increased viral transmission of Delta may help explain one mechanism of
how Delta became the predominant circulating variant (9). However, in agreeance with our
previous study, children infected with Delta still have a 45% reduction in the odds of having a
culture positive sample compared to adults (20).

Children have borne a significant burden from disruptions in developmental, social, and
educational aspects from the SARS-CoV-2 pandemic, all while their role in the spread of disease
has been continuously scrutinized. Although earlier studies suggested that the spread of disease
from and amongst children is low, more recent evidence suggests that this may be changing
(15,21). As highly transmissible VOCs, such as Delta, arise globally, there is a renewed concern
regarding the pediatric populations role in seeding outbreaks and an increasing importance to
understanding the infection dynamics in this population (2,22). In the absence of epidemiological
evidence of the spread of the various variants of SARS-CoV-2 amongst and from children, we
investigated differences in both Ct and viable virus levels in children infected with Delta
compared to OVs. We then further compared the pediatric population infection trends to the
adult population.

The physiological mechanism for age discrepancies for SARS-CoV-2 infection is still
widely under investigation due to the potentially large number of biological, host and
environmental factors (23). These factors may include differences in the dynamics of shedding

from children as compared to adults and/or reflect differences in maturity of their respective
immune systems (24).

While it has been generally accepted that higher secondary attack rates are associated
with higher viral loads as measured by quantitative nucleic acid amplification methods (i.e. RT-
PCR), others have shown that nucleic acid detection does not necessarily coincide with the
presence of infectious virus (20,25). This latter point is apparent in the results shown here
finding similar Ct levels in children infected with Delta and OV, however, **Delta positive**
**samples produced much higher culture positivity compared to previous variants.** Here culture
positivity was evaluated using TCID₅₀/mL titres on cells as a surrogate for transmissibility, a
much more specific test of infectivity than RT-PCR. This difference between Delta and OV
culture positivity in children may reveal one aspect of its transmissibility advantage (26).

Parents are encouraged to vaccinate as well as implement other non-pharmaceutical
public health interventions such as social distancing, staying home when sick, hand hygiene and
mask use. Some resistance to these measures has come from the fact that, on balance, children
generally do well following infection. The virological data showing increased infectious
potential of Delta VOC in pediatric individuals reiterates the importance of vaccination,
especially with the recent approval of the BioNTech COVID-19 vaccine for children 5-11 (26).
Recognizing this, it reinforces the need for a whole-of-society approach to vaccination and other
interventions, since the protection against severe outcome involves all.

**4.2. Limitations**

There are limitations of this study that should be considered. This includes the lack of
clinical and epidemiological data linked to samples from this study. Incorporation of this data is
an important next step in establishing SARS-CoV2 Delta VOC transmission dynamics in

children. Additionally, without longitudinal sampling any difference in onset and duration of
viral shedding between Delta and OVs in children cannot be determined. The duration of
infectivity likely plays a major role in increased viral transmission of Delta compared to OVs
and should be further investigated. It is also possible that differences in RT-PCR and culturable
virus between children and adults are accounted for by the nature or quality of the swab sampling
differences between these demographics. However, the fact that the quantification of the house-
keeping gene (BGB) from adults and children are comparable make this less likely (Figure 3).
Furthermore, the same samples are used for PCR testing and culturable virus and so if this is an
accepted critique, we should not consider similar Ct cut-offs in adults and children.

**4.3. Conclusion**

The relationship between age, SARS-CoV-2 viral load, and transmission has not been
comprehensively explored and requires further investigation. Continuation of this work also
involves monitoring of viral dynamics for new emerging SARS-CoV-2 VOCs, such as Omicron
(B.1.1.529) in different age populations. However, the results presented, showing Delta variant
grows more often and to higher levels from nasopharyngeal swabs taken from children compared
to previous variants of SARS-CoV-2, may have revealed changing infection trends within the
pediatric demographic with novel VOCs.

**References**

- 1. Dawood FS, Porucznik CA, Veguilla V, Stanford JB, Duque J, Rolfes MA, et al.
Incidence Rates, Household Infection Risk, and Clinical Characteristics of SARS-CoV-2
Infection among Children and Adults in Utah and New York City, New York. *JAMA*
*Pediatr.* 2021;30329:1–9.
- 2. Goldstein E, Lipsitch M, Cevik M. On the Effect of Age on the Transmission of SARS-
CoV-2 in Households, Schools, and the Community. *J Infect Dis* [Internet]. 2021 Feb
13;223(3):362–9. Available from: <https://academic.oup.com/jid/article/223/3/362/5943164>
- 3. Ladhani SN, Baawuah F, Beckmann J, Okike IO, Ahmad S, Garstang J, et al. SARS-CoV-
2 infection and transmission in primary schools in England in June–December, 2020
(sKIDS): an active, prospective surveillance study. *Lancet Child Adolesc Heal* [Internet].

- 2021 Jun;5(6):417–27. Available from:
<https://linkinghub.elsevier.com/retrieve/pii/S2352464221000614>
- 4. Soriano-Arandes A, Gatell A, Serrano P, Biosca M, Campillo F, Capdevila R, et al.
Household Severe Acute Respiratory Syndrome Coronavirus 2 Transmission and
Children: A Network Prospective Study. *Clin Infect Dis* [Internet]. 2021 Sep
15;73(6):e1261–9. Available from:
<https://academic.oup.com/cid/article/73/6/e1261/6168547>
- 5. Stein-Zamir C, Abramson N, Shoob H, Libal E, Bitan M, Cardash T, et al. A large
COVID-19 outbreak in a high school 10 days after schools’ reopening, Israel, May 2020.
*Eurosurveillance* [Internet]. 2020 Jul 23;25(29). Available from:
<https://www.eurosurveillance.org/content/10.2807/1560-7917.ES.2020.25.29.2001352>
- 6. Pray IW, Gibbons-Burgener SN, Rosenberg AZ, Cole D, Borenstein S, Bateman A, et al.
COVID-19 Outbreak at an Overnight Summer School Retreat — Wisconsin, July–August
2020. *MMWR Morb Mortal Wkly Rep* [Internet]. 2020 Oct 30;69(43):1600–4. Available
from: http://www.cdc.gov/mmwr/volumes/69/wr/mm6943a4.htm?s_cid=mm6943a4_w
- 7. Parri N, Magistà AM, Marchetti F, Cantoni B, Arrighini A, Romanengo M, et al.
Characteristic of COVID-19 infection in pediatric patients: early findings from two Italian
Pediatric Research Networks. *Eur J Pediatr* [Internet]. 2020 Aug 3;179(8):1315–23.
Available from: <https://link.springer.com/10.1007/s00431-020-03683-8>
- 8. Lam-Hine T, McCurdy SA, Santora L, Duncan L, Corbett-Detig R, Kapusinszky B, et al.
Outbreak Associated with SARS-CoV-2 B.1.617.2 (Delta) Variant in an Elementary
School - Marin County, California, May-June 2021. *MMWR Morb Mortal Wkly Rep*
[Internet]. 2021 Sep 3;70(35):1214–9. Available from:
<http://www.ncbi.nlm.nih.gov/pubmed/34473683>
- 9. GISAIID. Map of tracked variant occurrence [Internet]. 2021 [cited 2021 Dec 7]. Available
from: <https://www.gisaid.org/hcov19-variants/>
- 10. Luo CH, Morris CP, Sachithanandham J, Amadi A, Gaston D, Li M, et al. Infection with
the SARS-CoV-2 Delta Variant is Associated with Higher Infectious Virus Loads
Compared to the Alpha Variant in both Unvaccinated and Vaccinated Individuals.
*medRxiv Prepr Serv Heal Sci* [Internet]. 2021; Available from:
<http://www.ncbi.nlm.nih.gov/pubmed/34462756> [http://www.pubmedcentral.nih.gov/a](http://www.pubmedcentral.nih.gov/articlerender.fcgi?artid=PMC8404894)
[rticlerender.fcgi?artid=PMC8404894](http://www.pubmedcentral.nih.gov/articlerender.fcgi?artid=PMC8404894)
- 11. Planas D, Veyer D, Baidaliuk A, Staropoli I, Guivel-Benhassine F, Rajah MM, et al.
Reduced sensitivity of SARS-CoV-2 variant Delta to antibody neutralization. *Nature*
[Internet]. 2021 Aug 12;596(7871):276–80. Available from:
<https://www.nature.com/articles/s41586-021-03777-9>
- 12. dos Santos WG. Impact of virus genetic variability and host immunity for the success of
COVID-19 vaccines. *Biomed Pharmacother* [Internet]. 2021 Apr;136:111272. Available
from: <https://linkinghub.elsevier.com/retrieve/pii/S0753332221000573>
- 13. Lopez Bernal J, Andrews N, Gower C, Gallagher E, Simmons R, Thelwall S, et al.
Effectiveness of Covid-19 Vaccines against the B.1.617.2 (Delta) Variant. *N Engl J Med*
[Internet]. 2021 Aug 12;385(7):585–94. Available from:
<http://www.nejm.org/doi/10.1056/NEJMoa2108891>
- 14. Ong SWX, Chiew CJ, Ang LW, Mak T-M, Cui L, Toh MPHS, et al. Clinical and
Virological Features of Severe Acute Respiratory Syndrome Coronavirus 2 (SARS-CoV-
2) Variants of Concern: A Retrospective Cohort Study Comparing B.1.1.7 (Alpha),

- B.1.351 (Beta), and B.1.617.2 (Delta). *Clin Infect Dis* [Internet]. 2021 Aug 23; Available
from: <https://academic.oup.com/cid/advance-article/doi/10.1093/cid/ciab721/6356459>
- 15. Munoz FM. If Young Children’s Risk of SARS-CoV-2 Infection Is Similar to That of
Adults, Can Children Also Contribute to Household Transmission? *JAMA Pediatr*
[Internet]. 2021 Oct 8; Available from:
<https://jamanetwork.com/journals/jamapediatrics/fullarticle/2785008>
- 16. Delahoy MJ, Ujamaa D, Whitaker M, O’Halloran A, Anglin O, Burns E, et al.
Hospitalizations Associated with COVID-19 Among Children and Adolescents —
COVID-NET, 14 States, March 1, 2020–August 14, 2021. *MMWR Morb Mortal Wkly*
*Rep* [Internet]. 2021 Sep 10;70(36):1255–60. Available from:
http://www.cdc.gov/mmwr/volumes/70/wr/mm7036e2.htm?s_cid=mm7036e2_w
- 17. Freed NE, Vlková M, Faisal MB, Silander OK. Rapid and inexpensive whole-genome
sequencing of SARS-CoV-2 using 1200 bp tiled amplicons and Oxford Nanopore Rapid
Barcoding. *Biol Methods Protoc* [Internet]. 2020 Jan 1;5(1). Available from:
<https://academic.oup.com/biomethods/article/doi/10.1093/biomethods/bpaa014/5873518>
- 18. Moldovan E, Moldovan V. Controls in real-time polymerase chain reaction based
techniques. *Acta Marisiensis - Ser Medica*. 2020;66(3):79–82.
- 19. Ramakrishnan MA. Determination of 50% endpoint titer using a simple formula. *World J*
*Virol*. 2016;5(2):85–6.
- 20. Bullard J, Funk D, Dust K, Garnett L, Tran K, Bello A, et al. Infectivity of severe acute
respiratory syndrome coronavirus 2 in children compared with adults. *Can Med Assoc J*
[Internet]. 2021 Apr 26;193(17):E601–6. Available from:
<http://www.cmaj.ca/lookup/doi/10.1503/cmaj.210263>
- 21. Allen H, Vusirikala A, Flannagan J, Twohig KA, Zaidi A, Chudasama D, et al. Household
transmission of COVID-19 cases associated with SARS-CoV-2 delta variant (B.1.617.2):
national case-control study. *Lancet Reg Heal - Eur* [Internet]. 2022 Jan;12:100252.
Available from: <https://linkinghub.elsevier.com/retrieve/pii/S2666776221002386>
- 22. Zhu Y, Bloxham CJ, Hulme KD, Sinclair JE, Tong ZWM, Steele LE, et al. A Meta-
analysis on the Role of Children in Severe Acute Respiratory Syndrome Coronavirus 2 in
Household Transmission Clusters. *Clin Infect Dis* [Internet]. 2021 Jun 15;72(12):e1146–
53. Available from: <https://academic.oup.com/cid/article/72/12/e1146/6024998>
- 23. Mossong J, Hens N, Jit M, Beutels P, Auranen K, Mikolajczyk R, et al. Social Contacts
and Mixing Patterns Relevant to the Spread of Infectious Diseases. Riley S, editor. *PLoS*
*Med* [Internet]. 2008 Mar 25;5(3):e74. Available from:
<https://dx.plos.org/10.1371/journal.pmed.0050074>
- 24. Dong Y, Mo X, Hu Y, Qi X, Jiang F, Jiang Z, et al. Epidemiology of COVID-19 Among
Children in China. *Pediatrics* [Internet]. 2020 Jun;145(6):e20200702. Available from:
<https://publications.aap.org/pediatrics/article/76952>
- 25. Marc A, Kerioui M, Blanquart F, Bertrand J, Mitjà O, Corbacho-Monné M, et al.
Quantifying the relationship between SARS-CoV-2 viral load and infectiousness. *Elife*
[Internet]. 2021 Sep 27;10. Available from: <https://elifesciences.org/articles/69302>
- 26. Walter EB, Talaat KR, Sabharwal C, Gurtman A, Lockhart S, Paulsen GC, et al.
Evaluation of the BNT162b2 Covid-19 Vaccine in Children 5 to 11 Years of Age. *N Engl*
*J Med* [Internet]. 2021 Nov 9; Available from:
<http://www.nejm.org/doi/10.1056/NEJMoa2116298>

 Figure 1: Comparison of SARS-CoV-2 E gene RT-PCR cycle threshold (Ct) value by variant of
 concern vs. age group. Median Ct was higher for the Delta variant in pediatric age groups than
 for adults (P <0.001 Kruskal-Wallis ANOVA) implying lower viral RNA levels. Median [IQR]
 Ct was similar for children ages 0-17 between OV and Delta variants (24, [18-30] vs. 23, [20-
 28], P 0.97).

Figure 2: Comparison of Tissue Culture Infective dose 50% (TCID₅₀) per mL by variant of
 concern (VOC) vs. age group. There was no difference in TCID₅₀/ml between the different age
 categories with respect to the Delta variant specifically (P 0.68 Kruskal-Wallis ANOVA). In the
 pediatric age groups, the TCID₅₀/ml was significantly higher for the Delta variant in the 11-17
 age category than for OV (5620 [1780-17800] vs. 316 [178-2125], P <0.001), but there was no
 difference in the 0-10 age category.

Figure 3: Linear regression analysis of swab beta globin (BGB) Ct vs age shows no statistical
difference between BGB Ct level and age at which the sample was obtained.

Fig. 1

Fig. 2

Fig. 3

Responses to Reviewers

Reviewer 1:

Please delete subheadings labelling numbers in the manuscript e.g. 1.0. Introduction
All numbering format for subheadings has been removed.

Abstract:

The abstract is concise and explains the study well. However to be in line with the journal style, replace the heading "Interpretation" with "Importance".

The heading titled interpretation has been changed to Importance.

Methods:

SARS-CoV-2 RT-PCR

For the variant mutations, they were mentioned "N501Y, 484K and 452R", please write them consistently. E.g E484K etc

Thank you for pointing out this inconsistency, the single nucleotide polymorphism nomenclature has been updated on line 159 "N501Y, E484K and L452R"

Statistical analysis:

Page 6, line 180 -184: There should be citation of your previous work on this subject and citation of other studies where you mentioned about the delta variant infectious rate. In addition, what method was used to calculate the sample size, please elaborate on this "In our previous work, we found that adults had a culture positivity rate of 28.9%. Based on published data that the Delta variant may be 40-60% more infectious than wild type SARS-CoV-2, and assuming that this value would be related to culture positivity, we would require 116 pediatric samples to detect a 40% increase in culture positive rates, with a power of 0.8 and α of 0.05 among children."

We apologize for missing these references beforehand, the following references have been added as suggested:

21. Bullard J, Dust K, Funk D, Strong JE, Alexander D, Garnett L, Boodman C, Bello A, Hedley A, Schiffman Z, Doan K, Bastien N, Li Y, van Caesele PG, Poliquin G. 2020. Predicting infectious severe acute respiratory syndrome coronavirus 2 from diagnostic samples. Clin Infect Dis 71:2663–2666.
22. Yang W, Shaman J. 2022. COVID-19 pandemic dynamics in India, the SARS-CoV-2 Delta variant and implications for vaccination. J R Soc Interface 19.
23. Liu Y, Rocklöv J. 2021. The reproductive number of the Delta variant of SARS-CoV-2 is far higher compared to the ancestral SARS-CoV-2 virus. J Travel Med 28.

Figures

Write a figure legend page and the next pages put the figures without captions

A figure legend page has been added on page 13 with the figures following without the respective figure legends.

In page 1, line 377: Figure 1 please explain what does these numbers mean? "(24, [18-30] vs. 23, [20377 28], P 0.97)."

Thanks for pointing this out as it was not clear in the original. This is in response to the Figure 1 legend (now line 419-423) comparing the SARS-CoV-2 positive cycle threshold values by variant in children. The numbers in question represent the median cycle threshold and its associated interquartile range “(24, [18-30] vs. 23, [20-28], P=0.97)” for children aged 0-17 infected with the original variant and delta variant respectively.

To help clarify this we have added a sentence to the methods section under statistical analysis on line 184 showing a representation of how the statistical results are presented “Statistical results are reported as (Median [Interquartile range], P value).”. Similarly a simplified version “Median [IQR]” precedes the statement in question on Line 421.

In page 14, line 382: Figure 2 please write p-value as p= 0.68

This grammatical error has been corrected, thank you.

In page 14, line 384: explain what these numbers are?

"Figure 2: Comparison of Tissue Culture Infective dose 50% (TCID50) per mL by variant of concern (VOC) vs. age group. There was no difference in TCID50/ml between the different age categories with respect to the Delta variant specifically (P 0.68 Kruskal-Wallis ANOVA). In the pediatric age groups, the TCID50/ml was significantly higher for the Delta variant in the 11-17 age category than for OV (5620 [1780-17800] vs. 316 [178-2125], P <0.001), but there was no difference in the 0-10 age category."

The numbers in figure 2 caption are the median TCID50/mL and the associated interquartile range for children (aged 11-17) infected with delta variant (5620[1780-17800]) and the original variant (316[178-2125]). As these numbers showed a significant difference we have also included the P value.

For clarity we have added a sentence to the statistical methods section Line 184 depicting how the statistical results are presented throughout the paper. “Statistical results are reported as (Median [Interquartile range], P value).”

Reviewer 3:

In lines (118-120), You need updated statistical data for Coronavirus disease (COVID-19) pandemic.

The statistics on case and deaths has been updated as of July 15, 2022 on line 120. “Severe acute respiratory syndrome 2 (SARS-CoV-2), the etiological agent of COVID-19, has caused over 556 million cases and 6.3 million deaths globally as of July 15, 2022 (WHO Dashboard).”

In lines (239-243) It is preferable to delete this sentence. We still do not know what the side effects of vaccines are. Also, some studies have confirmed that there are side effects to the vaccine, and there are many opponents to the vaccine. Also, there are many voices that have not agreed to give vaccinations to children. The results of this study will tell us important things about how new mutations spread and how to stop them, and you don't need to mention the vaccine at the moment to children.

The sentence has been deleted as suggested.

In References, You need to standardize the format for all references (style). There are some errors in the year of publication, as well as the contents of some references.

Thank you for pointing this out. We have gone through the references and made it a consistent format as well as corrected the mistakes as follows:

1. Dawood FS, Porucznik CA, Veguilla V, Stanford JB, Duque J, Rolfes MA, Dixon A, Thind P, Hacker E, Castro MJE, Jeddy Z, Daugherty M, Altunkaynak K, Hunt DR, Kattel U, Meece J, Stockwell MS. 2022. Incidence Rates, Household Infection Risk, and Clinical Characteristics of SARS-CoV-2 Infection among Children and Adults in Utah and New York City, New York. *JAMA Pediatr* 30329:1–9.
*changed to 2022
10. Luo CH, Morris CP, Sachithanandham J, Amadi A, Gaston DC, Li M, Swanson NJ, Schwartz M, Klein EY, Pekosz A, Mostafa HH. 2021. Infection with the SARS-CoV-2 Delta Variant is Associated with Higher Recovery of Infectious Virus Compared to the Alpha Variant in both Unvaccinated and Vaccinated Individuals. *Clin Infect Dis*. 2021 Dec 18;ciab986.c doi: 10.1093/cid/ciab986.
*No longer a preprint- now published in *Clinical Infectious Diseases*
24. Allen H, Vusirikala A, Flannagan J, Twohig KA, Zaidi A, Chudasama D, Lamagni T, Groves N, Turner C, Rawlinson C, Lopez-Bernal J, Harris R, Charlett A, Dabrera G, Kall M. 2021. Household transmission of COVID-19 cases associated with SARS-CoV-2 delta variant (B.1.617.2): national case-control study. *Lancet Reg Heal - Eur* 12:100252.
*changed to 2021
29. Walter EB, Talaat KR, Sabharwal C, Gurtman A, Lockhart S, Paulsen GC, Barnett ED, Muñoz FM, Maldonado Y, Pahud BA, Domachowske JB, Simões EAF, Sarwar UN, Kitchin N, Cunliffe L, Rojo P, Kuchar E, Rämets M, Munjal I, Perez JL, Frenck RW, Lagkadinou E, Swanson KA, Ma H, Xu X, Koury K, Mather S, Belanger TJ, Cooper D, Türeci Ö, Dormitzer PR, Şahin U, Jansen KU, Gruber WC. 2022. Evaluation of the BNT162b2 Covid-19 Vaccine in Children 5 to 11 Years of Age. *N Engl J Med* 35–46.
*changed to 2022

August 1, 2022

Dr. James Eric Strong
Public Health Agency of Canada
Division of Special Pathogens
1015 Arlington Street
Winnipeg, Manitoba R3E 3P6
Canada

Re: Spectrum00395-22R1 (Differential infectivity of original and Delta variants of SARS-CoV-2 in children compared to adults)

Dear Dr. James Eric Strong:

Thank you for responding to the reviewers comments.

Your manuscript has been accepted, and I am forwarding it to the ASM Journals Department for publication. You will be notified when your proofs are ready to be viewed.

Sincerely,

Alison Sinclair
Editor, Microbiology Spectrum